# FLIM for Evaluation of Difference in Metabolic Status between Native and Differentiated from iPSCs Dermal Papilla Cells

**DOI:** 10.3390/cells11172730

**Published:** 2022-09-01

**Authors:** Alena Kashirina, Alena Gavrina, Artem Mozherov, Dmitriy Kozlov, Daria Kuznetsova, Ekaterina Vorotelyak, Elena Zagaynova, Ekaterina Kalabusheva, Aleksandra Kashina

**Affiliations:** 1Institute of Experimental Oncology and Biomedical Technologies, Privolzhsky Research Medical University, 603005 Nizhny Novgorod, Russia; 2Koltzov Institute of Developmental Biology, Russian Academy of Sciences, 119334 Moscow, Russia; 3Biology Department, Lomonosov Moscow State University, 119899 Moscow, Russia; 4Institute of Biology and Biomedicine, Lobachevsky State University of Nizhni Novgorod, 603022 Nizhny Novgorod, Russia

**Keywords:** iPSC, dermal spheroids, metabolism, NAD(P)H, pH_i_, BCECF, SypHer-2, FLIM

## Abstract

iPSCs and their derivatives are the most promising cell sources for creating skin equivalents. However, their properties are not fully understood. In addition, new approaches and parameters are needed for studying cells in 3D models without destroying their organization. Thus, the aim of our work was to study and compare the metabolic status and pH of dermal spheroids created from dermal papilla cells differentiated from pluripotent stem cells (iDP) and native dermal papilla cells (hDP) using fluorescence microscopy and fluorescence lifetime imaging microscopy (FLIM). For this purpose, fluorescence intensities of NAD(P)H and FAD, fluorescence lifetimes, and the contributions of NAD(P)H, as well as the fluorescence intensities of SypHer-2 and BCECF were measured. iDP in spheroids were characterized by a more glycolytic phenotype and alkaline intra-cellular pH in comparison with hDP cells. Moreover, the metabolic activity of iDP in spheroids depends on the source of stem cells from which they were obtained. So, less differentiated and condensed spheroids from iDP-iPSDP and iDP-iPSKYOU are characterized by a more glycolytic phenotype compared to dense spheroids from iDP-DYP0730 and iDP-hES. FLIM and fluorescent microscopy in combination with the metabolism and pH are promising tools for minimally invasive and long-term analyses of 3D models based on stem cells.

## 1. Introduction

Topical problems in dermatology encourage researchers to search for new approaches and techniques for solving. Creating skin equivalents is such an approach. Currently, both full-layer equivalents of the skin and its layers, which have living cells of various types and origins, are being created and studied. Of greatest interest is the creation of full-layer skin equivalents [1,2]. Most of the established skin equivalents are composed of primary keratinocytes and fibroblasts; however, other cell lines such as dermal papilla cells are used to improve models and create skin appendages [3].

Dermal papilla (DP) cells are mesenchymal cells with special functions, which are one of the main cell populations of the hair follicle (HF) that determine its viability and functional activity [4,5]. In addition, the assignment of DP cells to mesenchymal stem cells gives every reason to believe that skin equivalents containing DP cells will accelerate neoangiogenesis, remodeling of the extracellular matrix, the formation of granulation tissue, and skin regeneration [3]. However, the quality of skin equivalents and, in particular, the efficiency of experimental human HF regeneration is affected by both the developed approaches and the biological properties of donor cells [6,7,8,9]. DP cells obtained from hair follicles of patients with hair loss are limited in number. In addition, DP cells are known to lose their trichogenic properties during cell division when cultured in vitro [7,8,10]. To solve these problems, an alternative source of cells is needed.

The creation of iPSCs is a breakthrough in the field of regenerative medicine. Recent studies have shown that the use of iPSCs makes it possible to obtain effective tissue engineering products, including those for the treatment of skin wounds [11]. Protocols for the differentiation of iPSCs into keratinocytes [12,13], dermal fibroblasts [12], and dermal papilla cells [14,15] have already been successfully developed. The study of DP cells (iDP) differentiated from iPSCs has shown their differences from native DP cells. Even though iDP quite intensively expresses the genes associated with the generation of hair follicles compared to native DP cells and forms HF-like structures, however, HF differs in morphology and functioning from native ones [10,16]. These results indicated that further studies are needed to improve the methodology for obtaining DP cells from hiPSCs and generating skin equivalents with appendages.

Comparative analysis of iPSC-derived cells in three-dimensional structures with native cells cultivated under similar conditions is another topical task of regenerative medicine. Traditionally, this analysis is carried out on the expression of characteristic genes and surface markers using ICC (immunocytochemistry) and PCR (polymerase chain reaction). However, the constant evolution of optical imaging methods allows expanding the horizons of research. Thus, fluorescence microscopy in combination with FLIM (fluorescence lifetime imaging microscopy) makes it possible to study three-dimensional cell models both in vitro and in vivo, after transplantation, with less invasiveness, high sensitivity, and specificity, in real-time, as well as to monitor processes on the same models. FLIM of endogenous cofactors NAD(P)H and FAD/FMN provides information about the relative changes in the activity of cellular energy-producing pathways [17,18,19]. The relevance of the analysis of energy metabolism is due to its direct influence on the functioning of the cell, including the cell cycle, synthesis, breakdown, and transport of substances. The difference in the functioning of different cells of the body is also reflected in the difference in energy metabolism, which in turn is expressed in differences in the lifetimes and amplitudes of NADH and flavins. It is important that the physiological functioning of cells in three-dimensional models is greatly influenced by the microenvironment. Metabolic imaging with FLIM makes it possible to study cells in real time while interacting with a specific microenvironment, both with other cell lines and with a matrix. Thus, energy metabolism as measured by FLIM can be used as a sensitive parameter for comparative analysis of stem cell models with native cell models.

The switching of metabolic pathways is directly related to changes in intracellular pH. More alkaline pH values are known to promote glycolysis in cells [20]. Fluorescence microscopy combined with genetically encoded sensors allows the analysis of pH changes with a minimal invasive impact on samples. Thus, fluorescence microscopy and FLIM in combination with endogenous fluorophores and genetically encoded sensors make it possible to use the features of energy metabolism and intracellular pH as markers in the comparative analysis of tissue-engineered skin constructs obtained from iPSCs with those from native human cells in dynamics and with minimal invasive impact.

This work is aimed to study and analyze the metabolic status and pH of dermal spheroids created from dermal papilla cells differentiated from PSC and native dermal papilla cells using fluorescence microscopy and FLIM.

## 2. Materials and Methods

### 2.1. Cell Culture and Differentiation

Dermal papilla cells (iDP) differentiated from human pluripotent stem cells (iPS-KYOU (ACS-1023™, ATCC, USA), iPS-DYP0730 (ACS-1023™, ATCC, USA), iPSC-DP [21], hES-MK05 (Vavilov Institute of General Genetics, Russian Academy of Sciences) and primary human dermal papilla cells (hDP) from 3 donors were used to create 3D dermal cultures. All the cell lines were deposited in the Cell culture collection of the Institute of developmental biology RAS. For culturing PSCs, the wells of culture plates were coated with sorbed Matrigel (1/60 in DMEM/F12 (cat #11320033)) (cat #354230) (BD Bioscience, Billerica, MA, USA). iPSCs were cultured in mTeSR1 medium (cat #85850) (Stem Cell Technologies, Vancouver, BC, Canada) at 37 °C in a CO_2_ incubator with 5% CO_2_ and 5% O_2_. The cells were passaged with 0.1% dispase; during passaging, 10 μM of the ROCK inhibitor Y-27632 (cat #72302) (Sigma Aldrich, Bulington, MA, USA) was added.

The differentiation protocol was developed within the framework of the Russian Science Foundation Grant No. 16-14-00204 and reproduces the morphogenetic processes of the formation of the dermal papilla of the hair follicle from the neural crest [14]. iPSCs and hESCs were preliminarily differentiated towards neural progenitor cells. Briefly, differentiation was initiated using the Neural Induction Kit (cat #A1647801) (Thermofisher, Waltham, MA, USA) on a plastic coated with Matrigel for 14 days. Further differentiation was induced using DMEM/F12 medium supplemented with insulin/transferrin/selenite (cat #41400045) (ITS) (Gibco, Thermo Fisher Scientific, Waltham, MA, USA) 10% FBS (cat #SH30071.03) (HyClone, Logan, UT, USA) and gentamicin/amphotericin (cat #R01510) (Gibco, Thermo Fisher Scientific, Waltham, MA, USA) (1/500) (Figure 1). After 3 weeks, the cell culture was transferred to an atmosphere of 21% O_2_.

### 2.2. Isolation of Primary Dermal Cells

Human scalp biopsies were obtained after face-lift surgery from informed and consented patients aged 49, 57, and 66 from Federal State Autonomous Institution “National Medical Research Center “Medical and Rehabilitation Center” of the Ministry of Health of the Russian Federation. Before cell isolation, skin samples were washed with Hank’s solution (cat #P021п) (PanEko, Moscow, Russia) with gentamicin.

The skin was incubated in 0.5% dispase II (cat #17105041) (Gibco, Thermo Fisher Scientific, Waltham, MA, USA) at 4 °C overnight. Subcutaneous fat with hair follicle bulbs was separated manually by surgical scissors and incubated in 0.2% collagenase type I (cat #17100017) (Gibco, Thermo Fisher Scientific, Waltham, MA, USA) for 2–3 h at 37 °C. In the next stage, HF bulbs were separated from fat by pipetting and centrifuging. To obtain hDPs, HF bulbs were additionally incubated in 0.2% collagenase type I for 3–4 h at 37 °C. hDP cells were purified by a series of low-speed centrifuges. The cells were cultured in AmnioMAX™-II medium (cat #11269016) (Gibco, Thermo Fisher Scientific, Waltham, MA, USA). Cells in passages 1–4 were used for all experiments if another one is not indicated.

### 2.3. Obtaining of Dermal Spheroids

iDP and hDP spheroids were obtained by the “hanging drop” cultivation method. Drops containing 10^4^ cells in 20 μL were placed on the lid of a Petri dish. Hanks’ solution was added to the bottom to prevent drying out. The cells were cultured for 3 days until the formation of spheroids, then they were transferred to low-adhesion plates (Corning, New York, USA). Dermal spheroids were cultured in DMEM/F12 medium supplemented with ITS, 10% FBS, and gentamicin/amphotericin (1/500) at 37 °C in an atmosphere of 5% CO_2_ and 21% O_2_. For imaging analysis, spheroids were transferred on the glass-bottom FluoroDishes coated with Matrigel in 0.5 mL DMEM media (Gibco, Thermo Fisher Scientific, Waltham, MA, USA) without phenol red and allowed to attach for 4–5 h.

### 2.4. Immunofluorescent Staining

To stain the monolayer cell cultures (iPSCs, NPCs, iDP, hDP cells) and dermal spheroids, the samples were fixed in 4% PFA (cat #P6148-500G) (Sigma-Aldrich, Burlington, MA, USA) at room temperature (22–24 °C) for 60 min. Then the cultures were incubated for 60 min in a blocking solution containing 1% Triton X-100, 1% Tween-20, and 5% BSA. Next, the spheroids were incubated with primary antibodies overnight at 4 °C. Following primary antibody were used: anti-Oct4 (Millipore, Burlington, MA, USA (cat # MAB4401)), anti-SOX2 (Millipore, Burlington, MA, USA (cat #MAB2018)), anti-TUBB3 (Millipore, Burlington, MA, USA (cat #MAB1637), anti-CD44 (Abcam, Cambridge, UK (cat #ab119348), 1:100); anti-αSMA (Abcam, Cambridge, UK (cat #ab5694), 1:200); anti-vimentin (Abcam, Cambridge, UK (cat #ab24525), 1:300); anti-fibronectin (Abcam, Cambridge, UK (cat #ab2413), 1:200); anti-versican (R&D systems, Minneapolis, MN, USA (cat #AF3054), 1:20); anti-Ki67 (Abcam, Cambridge, UK (cat #ab16667), 1:50). After that, they were washed with a blocking solution for a day. After 24 h, the samples were incubated with secondary antibodies (Alexa Fluor™ 546 Donkey anti-Rabbit IgG (cat #A10040), Alexa Fluor™ 488 Goat anti-Rat IgG (cat #A-11006), Alexa Fluor™ 488 Donkey anti-Goat IgG (cat #A-11055), Invitrogen (Waltham, MA, USA), and Alexa Goat anti-Chicken IgY H&L Alexa Fluor^®^ 488 (cat #A-11039), Abcam (Cambridge, UK) diluted 1:500 in a blocking solution overnight at 4 °C. The nuclei were contrasted with DAPI (cat #40011) (Biotium, Fremont, CA, USA). For imaging, cells were embedded in a supersaturated fructose solution.

### 2.5. Isolation of RNA and Quantitative PCR Analysis

#### 2.5.1. Isolation of RNA and Quantitative PCR Analysis of the Expression of Specialized Genes

Total RNA was isolated using the RNeasy Mini Kit (cat #74104) (Qiagen, Hilden, Germany). RNA concentration was measured using an Eppendorf BioPhotometer plus. For the synthesis of cDNA, 1 μg of RNA was used. Genomic DNA elimination and first-strand cDNA synthesis were conducted using the QuantiTect Reverse Transcription kit (cat #205311) (Qiagen, Hilden, Germany). Real-time PCR was performed using a LightCyclerr 96 (Roche, Basel, Switzerland). The temperature profile of cycles: (1) 95 °C for 10 min, (2) 40 cycles of 95 °C for 10 s and 60 °C for 40 s, (3) melt curve analysis with measurements between 60 °C and 95 °C. To prepare the reaction mixtures, a mixture of HS-SYBR + ROX (cat #PK156S, #PK156L) (Evrogen, Moscow, Russia) was used. Quantitative RT-PCR analysis was performed on three biological and three technical replicates. Household genes GAPDH, PSMB4, REEP5, C1ORF43, and ACTB were used for normalization. In further calculations, the ^ΔΔC^ method was used. The primer’s sequences are presented in Table 1.

#### 2.5.2. Quantitative PCR Analysis of the Expression of Metabolic Genes

Real-time PCR was performed using a CFX96 Real-Time PCR system (Applied Biosystems, Waltham, MA, USA) using SYBR Green dye-based PCR amplification assay. The PCR reaction contained 1-x GeneAmp PCR Buffer I (cat #10219244) (Applied Biosystems, Waltham, MA, USA), 250 μM of each dNTP, 0.5 nM of each primer (primer sequences are shown in Table 2), and 1 U of Taq M polymerase (cat #756-50) (Intifica, Saint Petersburg, Russia); total concentration of Mg^2+^ in the reaction was 3 mM and the reaction volume was 20 μL. The temperature profile of cycles: (1) 95 °C for 10 min (enzyme activation step); (2) 35 cycles of 95 °C for 15 s, 60 °C for 30 s, and 72 °C for 30 s; (3) hybridization 1 min 95 °C and 1 min 40 °C; (4) melt curve analysis with measurements between 60 °C and 95 °C. The reaction efficiency was determined by the calibration curve method. Quantitative RT-PCR analysis was performed using the CFX Maestro 2.3 software. The selection of reference genes was carried out using an integrated geNorm algorithm. The following reference genes were used: ABL1 and EIf2B1.

### 2.6. Analysis of Dermal Spheroid Energy Metabolism by Multiphoton Fluorescence Microscopy and FLIM

Fluorescence and time-resolved images were obtained using an LSM 880 (Carl Zeiss, Jena, Germany) equipped with a short-pulse femtosecond Ti: Sa laser Mai Tai HP with a pulse repetition rate of 80 MHz, and a duration of 140 ± 20 fs (Spectra-Physics, Milpitas, CA, USA) and a FLIM system for time-resolved microscopy (Becker&Hickle GmbH, Berlin, Germany). Fluorescent images of NAD(P)H were obtained with two-photon excitation of fluorescence at a wavelength of 750 nm, fluorescence was received in the range of 455–500 nm. Fluorescence images of FAD were obtained from pixels of at least 5000; all studies were carried out under constant conditions (37 °C and 5% CO_2_). Next, we calculated the optical redox ratio (ORR = IFAD/INAD(P)H) using ImageJ 1.52p software (NIH, Bethesda, MD, USA). Using SPCImage software (Becker&Hickle GmbH, Berlin, Germany) we analyzed FLIM images and registered the following parameters: the fluorescence lifetimes (τ1, τ2 (ps)) and the lifetimes’ contributions (α1, α2 (%)).

### 2.7. Analysis of Dermal Spheroid Intracellular pH Using Multiphoton Fluorescence Microscopy

In these experiments, to create iDP-iPSKYOU spheroids, cells stably transfected with SypHer-2, obtained earlier [22], were used. For 3D models from primary DP cells and differentiated DP cell lines (iDP-DYP0730, iDP-iPSDP and iDP-hES), BCECF (cat #B1170) (Invitrogen, Waltham, Massachusetts, USA) staining with a ratiometric pH dye was used. In preliminary experiments, calibrations were performed with SypHer-2 and BCECF according to the previously described method [20]. Following the data, calibration curves were constructed. Before imaging of iDP-iPSKYOU, the spheroids were transferred onto FluoroDish dishes (WPI, Sarasota, FL, USA), coated with Matrigel with the presence of the FluoroBrite medium (cat #15291866) (Gibco, Thermo Fisher Scientific, Waltham, MA, USA) for 4–5 h for adhesion. Before staining with BCECF (Thermo Fisher Scientific, Waltham, MA, USA), a stock solution of the 1 mM dye in DMSO was prepared. The final solution was obtained by 100-fold dilution in a culture medium. Before staining, the spheroids were transferred to 35 mm glass-bottom dishes coated with Matrigel for 4–5 h for adhesion. Next, the spheroids were washed with PBS and incubated in a culture medium containing a dye for 60 min at 37 °C with constant shaking (for better passage of the dye). After that, the dye was washed 1–2 times with PBS e and embedded in a FluoroBrite medium. Fluorescence images were obtained using an LSM 880 confocal fluorescence microscope (Carl Zeiss, Jena, Germany). For SypHer-2 and BCECF, fluorescence was excited at 405 nm and 488 nm, followed by detection in the 500–550 nm range. The resulting images were processed using the ImageJ software (NIH, Bethesda, MD, USA) and the ratio of fluorescence intensities (I488/I405 for SypHer-2 and BCECF) was calculated.

### 2.8. Statistical Analysis

For this research into the metabolic status, the images of 5 spheroids of each type and about 50 ROIs were analyzed. For pHi analysis, from 5 to 8 spheroids of each type and at least 50 ROI were analyzed. The results were processed and statistically analyzed using EXCEL (Redmond, Washington, USA), STATISTICA 12 (Tulsa, OK, USA), and GraphPad Prism 9 (San Diego, CA, USA). The redox ratio, FLIM data, and I488/I405 ratio were expressed as the mean values ± SD. The distribution of all data was first checked for normality using the Kolmogorov–Smirnov test and the Shapiro–Wilk test. The statistical significance of the differences was analyzed using the Mann–Whitney U test.

## 3. Results

### 3.1. Dermal Differentiation, and Formation of hDP and iDP Spheroids

Dermal papilla cells were successfully differentiated from pluripotent stem cells. It was confirmed by ICC and quantitative PCR (Figure 2 and Figure 3).

The cells of the primary and differentiated dermal papilla formed spheroids in the hanging drop model system. hDP spheroids had spherical form and cells were densely packed. The spheroids from iDP differed in morphology. Spheroids from cells derived from the hES and iPS-DYP0730 lines were closest in morphology to spheroids formed by primary cells, while iDP-iPSDP and iDP-iPSKYOU formed less condensed spheroids with a less smooth outer boundary.

We analyzed the expression of specialized markers using confocal microscopy (Figure 4). We detected the expression of the VERSICAN, CD44, VIMENTIN, and FIBRONECTIN and did not observe the expression of smooth muscle actin. VERSICAN was distributed over the entire thickness of the spheroids, while other mesenchymal markers were located in the superficial layer. There were single cells positive for ki67 closer to the outer border of the spheroids. The denser spheroids from the iDP-iPSDYP0730 and iDP-hES lines were closest to hDP spheroids in terms of marker expression pattern and the number of proliferating cells. Spheroids from iDP-iPSDP and iDP-iPSKYOU cells positive for CD44, VIMENTIN, FIBRONECTIN, and proliferation marker ki67 were found throughout the thickness of the spheroids.

Quantitative PCR analysis did not reveal significant differences in the expression of *VERSICAN*, *ALCALINE PHOSPHOTASE* (*ALP*), *VIMENTIN*, *FIBRONECTIN*, *NESTIN*, *Collagen I*; *Collagen III* using the Mann–Whitney U test (Figure 5). We observed several scatter in the expression of these markers both among primary cell lines and among cell lines obtained by differentiation of pluripotent cells. We assessed whether there are differences between the iDP lines that form more and less dense spheroids. The iDP-iPSDYP0730 and iDP-hES lines, which form denser spheroids, were characterized by low expression of *VERSICAN* and *NESTIN*, comparable to primary cells, but higher expression of *FIBRONECTIN*. In combination with a low level of proliferation in spheroids, as in the case of primary cells, the nature of their physiology suggests a more differentiated state of the iDP-iPSDYP0730 and iDP-hES lines, in comparison with the iDP-iPSDP and iDP-iPSKYOU lines.

### 3.2. Evaluation of hDP and iDP Spheroids Energy Metabolism Using FLIM

The contribution of the main metabolic pathways (glycolysis and OxPhos) to the total energy metabolism of cells in three-dimensional dermal models was assessed by analysis of the ratio of the fluorescence intensity of FAD to NAD(P)H (optical redox ratio (ORR)), the fluorescence lifetimes (τ1, τ2) and the contributions (α1, α2) of free and bound forms of NAD(P)H using the fluorescence microscopy and FLIM. Evaluation of ORR showed that the mean values of this indicator in all types of hDP and iDP spheroids do not have a statistically significant difference (Figure 6, Table 3). This suggests that cells in spheroids of all types have approximately the same level of metabolic activity, but does not indicate the contribution of metabolic pathways to the energy metabolism of cells. FLIM was used to solve this problem.

Analysis of the FLIM data showed no statistically significant differences in the fluorescence lifetimes (τm, τ1, τ2) and the contribution of the bound form of NAD(P)H (α2) between native DP cells from three donors at the age of 49, 57, and 66 years. An analysis of τm, τ1, and τ2 parameters between different iDP lines also did not reveal a statistically significant difference but revealed significant differences in α2. In general, all hDP lines were characterized by higher values of τm and α2 than iDP (Figure 7a–c, Table 3). Lower values of α2 in spheroids from all types of differentiated cell lines indicate the predominance of glycolysis in energy metabolism compared to spheroids from native cells. It is important to note the statistically significant difference in α2 between the iDP spheroids, with lower parameter values in the iDP-iPSDP and iDP-iPSKYOU compared to the iDP-iPSDYP0730 and iDP-hES lines (Figure 7d, Table 3). This result may indicate a more glycolytic status of iDP-iPSDP and iDP-iPSKYOU cells in the 3D model and more OxPhos metabolism (approximate to hDP) in iDP-iPSDYP0730 and iDP-hES. In addition, the obtained result is consistent with the morphological features of iDP spheroids and with the results of immunocytochemical and PCR studies. It is known that actively proliferating cells, like stem cells and tumor cells, rely predominantly on glycolysis for energy metabolism [18]. iDP-iPSDP and iDP-iPSKYOU in spheroids are less differentiated cells, as evidenced by an increased level of expression of proliferation markers—NESTIN and ki67, as well as a lower aggregation ability of these cells, which manifests itself in a less condensed state of spheroids.

### 3.3. Evaluation of hDP and iDP Spheroids Energy Metabolism Using Real-Time PCR

For a more detailed study of metabolic characteristics hDP and iDP spheroids the level of expression of genes encoding the main metabolic enzymes in the cell was analyzed. Namely, HK1, HK2, and LDHA encoding proteins that are localized in the cytoplasm and are associated with glycolysis. PDK1 and OGDH encoding proteins that are localized in mitochondria and are associated with OxPhos, as well as G6PD, encoding a protein involved in the pentose phosphate pathway (PPP). The analysis showed that, in general, iDP cells have higher levels of gene expression of glycolysis, OxPhos, and PPP. Statistical analysis of the data revealed that in iDP cells, the expression of HK1, HK2, and OGDH genes is higher (*p* ≤ 0.05) (Figure 8) than in hDP cells. However, the increase in the expression of HK1 and HK2 is more pronounced. The obtained data may indicate that despite the severity of all metabolic pathways in iDP cells, the expression of genes involved in glycolysis processes prevails. This result correlates well with the results obtained by FLIM and ICC and confirms the predominance of the glycolysis contribution to the overall metabolism of the *iDP* cells.

### 3.4. Evaluation of hDP and iDP Spheroids Intracellular pH Using Fluorescence Microscopy

In preliminary experiments, two calibrations were carried out with the pH sensors SypHer-2 and BCECF to convert conventional pH units to absolute units (Figure 9a,b).

Next, we analyzed the fluorescence intensities ratio for BCECF and SypHer-2 in all types of hDP and iDP spheroids. In the hDP spheroids, we have shown the absence of statistically significant differences in the ratio of fluorescence intensities for BCECF, taking values 1.79 ± 0.06, 1.79 ± 0.05, and 1.77 ± 0.05. These mean values, according to the calibration curve, corresponded to the intracellular pH: 7.35 ± 0.06, 7.35 ± 0.04, 7.35 ± 0.05 for hDP-d134, hDP-d144 and hDP-d192 spheroids, respectively (Figure 10). No statistically significant differences in the ratio of fluorescence intensities for SypHer-2 and BCECF were found also between iDP spheroids. The mean values I488/I405 were 0.62 ± 0.07, 0.6 ± 0.06, 0.59 ± 0.06, and 0.64 ± 0.04 which correspond to the intracellular pH: 7.56 ± 015, 7.51 ± 0.12, 7.51 ± 0.13, 7.61 ± 0.09 for iDP-iPSDP, iDP-iPSDYP0730, iDP-hES, and iDP-iPSKYOU spheroids, respectively (Figure 11). In general, when comparing the pH in iDP and hDP, iDP pH tends to shift towards the alkaline side. Thus, the more glycolytic status of iDP cells in spheroids correlates with the more alkaline intracellular pH compared to hDP cells in spheroids.

## 4. Discussion

In this study, we compared for the first time 3D cultures from DP cells differentiated from PSCs to native human cell models by analyzing energy metabolism and intracellular pH using minimally invasive fluorescent and FLIM methods.

The interest of scientists in developing approaches for obtaining dermal papilla cells from iPSCs is due to the loss of biological properties by native human DP cells during in vitro cultivation [10,23,24]. Under in vivo conditions, dermal papilla cells are in contact with a certain microenvironment in the form of an extracellular matrix and other cells such as hair follicle stem cells. Early studies show that culturing DP cells in 3D models similar to in vivo conditions promotes follicle formation [25]. Moreover, different iPSC lines have different propensity to differentiate towards DP-like cells. Thus, iPSCs obtained from normal human B7 fibroblasts, differentiated in the direction of DP cells, were not able to induce the formation of hair follicles during transplantation and had low trichogenic activity [14]. WD39-derived iDPs moderately expressed DP markers and showed low staining of the cytoplasmic membrane, which was also distinguished from DP [15]. These results emphasize the importance of evaluating the biological properties of DP lines differentiated from iPSCs in comparison with native DP cells in 3D models.

It is well known that cellular metabolism is one of the crucial factors to regulate the functioning of cells. The glycolytic metabolism of human dermal papilla cells has previously been shown to promote hair follicle formation. Namely, gene analysis showed that spheroids from DP are characterized by higher glucose metabolism and high activity of aerobic glycolysis, in comparison with monolayer culture. Both processes are necessary for the normal expression of genes associated with hair induction [26].

Today, studies of the metabolic activity of cells are carried out using both traditional methods (ICC, PCR, biochemical methods, etc.) and fluorescent methods of analysis (fluorescence and time-resolved microscopy). However, each of the methods individually has both advantages and disadvantages. For instance, the traditional methods are characterized by the complexity and duration of sample preparation, and cell fixation, which makes the study end-point. In addition to that, they allow us to study cells either in a monolayer or in suspension and are not able to reflect the behavior of cells and their parameters, taking into account a certain microenvironment that exists in 3D cell models.

FLIM of endogenous fluorophores has proven itself as a sensitive, minimally invasive method for assessing changes in the activity of ATP-producing pathways of cells in vitro and in vivo [27,28,29]. Importantly, the use of endogenous fluorescence allows non-invasive metabolic imaging of cells and tissues in their natural physiological microenvironment without disturbances associated with fixatives and contrast agents and without lengthy sample preparation [30,31]. Using FLIM, Meleshina and co-workers showed that energy metabolism changes during long-term cultivation of dermal equivalents containing mouse dermal papilla cells [32]. FLIM is also used to study other skin cells and tissue-engineered skin equivalents [33,34,35]. However, the limitations of fluorescence microscopy and FLIM include the difficulty of interpreting the results obtained. Therefore, it is necessary to carry out complex studies of three-dimensional cellular models in order to correctly and fully interpret the results. Moreover, the use of several approaches helps gain a deeper understanding of the metabolic features of studied cellular systems. In current work fluorescence and time-resolved microscopies, PCR analysis was used to study the energy metabolism and, associated with it, the level of intracellular pH of the obtained three-dimensional cell cultures.

Dermal spheroids were obtained from DP cells differentiated from four lines of the pluripotent stem cells (iPS-DP, iPS-DYP0730, iPS-KYOU, and hES) and native DP cells from three donors. Using FLIM, we demonstrated that the average values of τm and α2 across all lines of native DP cells in spheroids were higher than the average values of parameters across all lines of differentiated DP cells. These data indicate a more glycolytic status of iDP spheroids. This result was confirmed by PCR analysis of metabolic gene expression. Namely, in iDP cells, we found a predominance of the glycolysis contribution to the overall metabolism. Next, an analysis was carried out within the groups of hDP and iDP spheroids. hDP cells from donors with different ages in 3D models did not differ in FLIM parameters, which indicates a relatively similar level of metabolic activity. When analyzing the iDP spheroids, differences in α2 were found between different lines of iDP cells, namely, iDP-iPSDP and iDP-iPSKYOU lines were characterized by lower values of this parameter (*p* ≤ 0.05) which indicates their more glycolytic status. It should also be noted that iDP-iPSDP and iDP-iPSKYOU are characterized by a more pronounced expression of glycolysis genes compared to iPSDYP0730 and iDP-hES. The result obtained using FLIM correlates well with the morphological features of these spheroids. The iDP spheroids created from iDP-iPSDP and iDP-iPSKYOU were less condensed with an uneven edge. At the same time, iDP spheroids created from iPS-DYP0730 and hES were more similar in morphology to spheroids from human DP and were characterized by a denser structure and a smooth edge. For dermal papilla cells, a characteristic feature is the ability to aggregate [7,25]. The loose iDP spheroids may have resulted from the correspondence of iPSC-derived dermal papilla cells to cells of the embryonic dermis rather than mature skin. The recent study showed that the quality of differentiated DP cells and their ability to aggregate depends on the type of pluripotent stem cells. It was found that 201B7-hiPSC or RPC-hiPS771-2-derived iDP cells gave rise to cells that formed less compactly or irregularly aggregated structures [15]. In our study, we showed that iDP cells in spheroids derived from the iPSDP and iPSKYOU lines are less mature (differentiated) cells with glycolytic metabolism, which also form less condensed structures.

It is a known fact that the maintenance of intracellular pH at a certain level is directly related to certain features of cellular metabolism. A more alkaline pH has been shown to promote glycolysis in cells by increasing enzyme activity [36,37]. In an early work, we analyzed the change in intracellular pH during the differentiation of iPSCs into dermal papilla cells in a monolayer. We have shown that in the process of differentiation, intracellular pH is acidified by 0.2 units, which is explained by the active synthesis of collagen by cells [22]. Similar studies have not been carried out anymore; however, ratiometric pH sensors have been used to analyze the intracellular pH of human fibroblasts [38] and keratinocytes [39]. Therefore, here we also applied the pH measurement method, which reflects the situation in 3D conditions. For this purpose, multiphoton microscopy of pH-sensitive sensors BCECF and SypHer-2 was used. We have shown that hDP spheroids do not differ in intracellular pH between different native cell lines. A similar result was obtained for different iDP cell lines. However, we found differences in intracellular pH between hDP and iDP spheroids (*p* ≤ 0.05). The intracellular pH of differentiated DP cells in spheroids takes on more alkaline values which correlate well with their more glycolytic status. The pH level can act as an additional parameter for assessing the metabolic state of cells in 3D models.

## 5. Conclusions

In our study, a comprehensive analysis of the energy metabolism and intracellular pH in hDP and iDP spheroids was carried out using fluorescence microscopy and FLIM. We have shown for the first time that spheroids from differentiated dermal papilla cells are characterized by a more glycolytic status compared to spheroids from native cells. PCR analysis of metabolic gene expression confirms the predominance of the glycolysis contribution to the overall metabolism of the iDP cells. The predominance of glycolysis in the energy metabolism of iDP spheroids correlates with more alkaline values of intracellular pH in these cells. Analysis of hDP and iDP spheroids within the groups showed that native DP cells from different donors have a similar level of metabolic activity. In iDP spheroids, it was found that less condensed spheroids from iDP-iPSDP and iDP-iPSKYOU are characterized by a more glycolytic phenotype compared to dense spheroids from iDP-DYP0730 and iDP-hES. So, the metabolic activity of differentiated cells in dermal spheroids depends on the source of stem cells from which they were obtained. The result obtained using FLIM correlates well with the level of expression of character markers and genes, which indicates a less differentiated state of iDP-iPSDP and iDP-iPSKYOU compared to iDP-DYP0730 and iDP-hES, as well as with the level of expression of metabolic genes, confirms the predominance of glycolysis in iDP-iPSDP and iDP-iPSKYOU cell lines. Thus, FLIM in combination with NAD(P)H is an effective approach for the metabolic analysis of various cell models without staining and destruction and in combination with pH investigation, promising tools for evaluating stem cell-based 3D systems in regenerative medicine.

## Figures and Tables

**Figure 1 cells-11-02730-f001:**
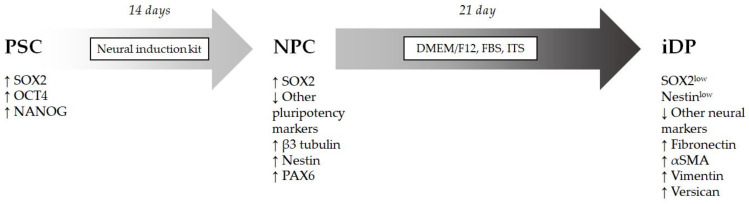
Differentiation of pluripotent stem cells into dermal papilla cells.

**Figure 2 cells-11-02730-f002:**
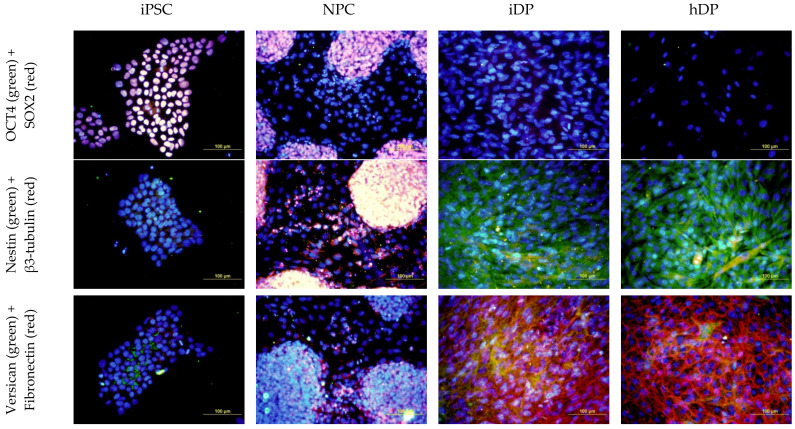
Analysis of the expression of specialized markers in iPSC, NPC, iDP, and hDP cell lines: OCT4 (green), SOX2 (red); NESTIN (green), β3-TUBULIN (red); VERSICAN (green), FIBRONECTIN (red). Fluorescence microscopy, the scale length in all pictures is 100 µm.

**Figure 3 cells-11-02730-f003:**
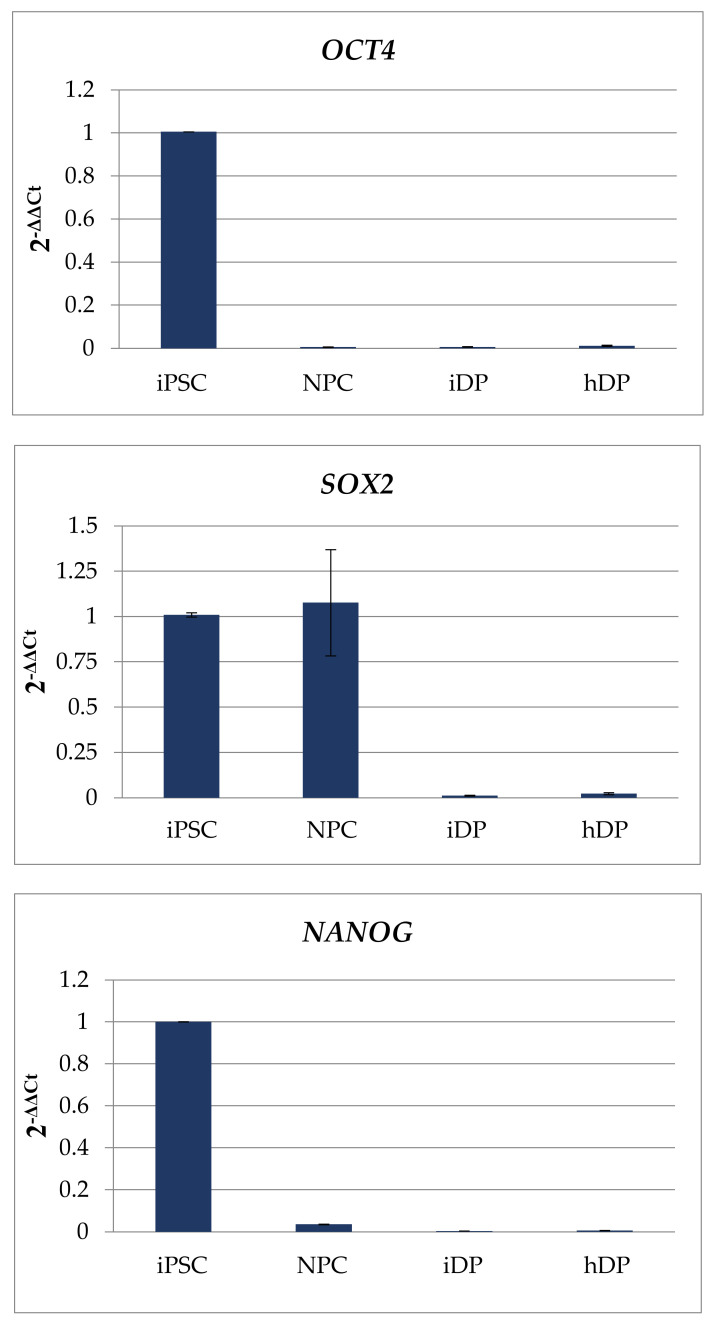
Expression of specialized markers in iPSC, NPC, iDP, and hDP cell lines. Quantitative PCR analysis: *OCT4, SOX2, NANOG, PAX6, NESTIN, β3-TUBULIN, VIMENTIN, FIBRONECTIN,*
*αSMA*.

**Figure 4 cells-11-02730-f004:**
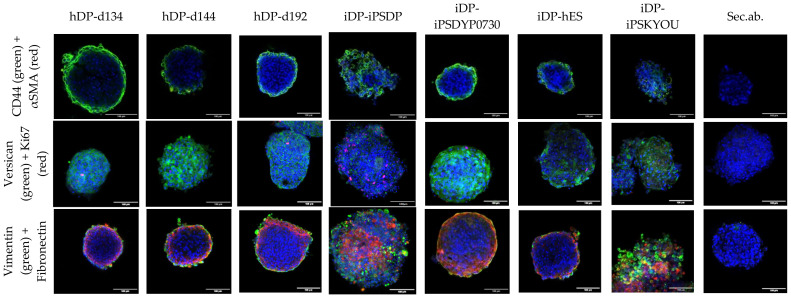
Analysis of the expression of specialized markers in the dermal spheroids: CD44 (green), αSMA (red); VERSICAN (green), Ki67 (red); VIMENTIN (green), FIBRONECTIN (red). Fluorescence microscopy, the scale length in all pictures is 100 µm.

**Figure 5 cells-11-02730-f005:**
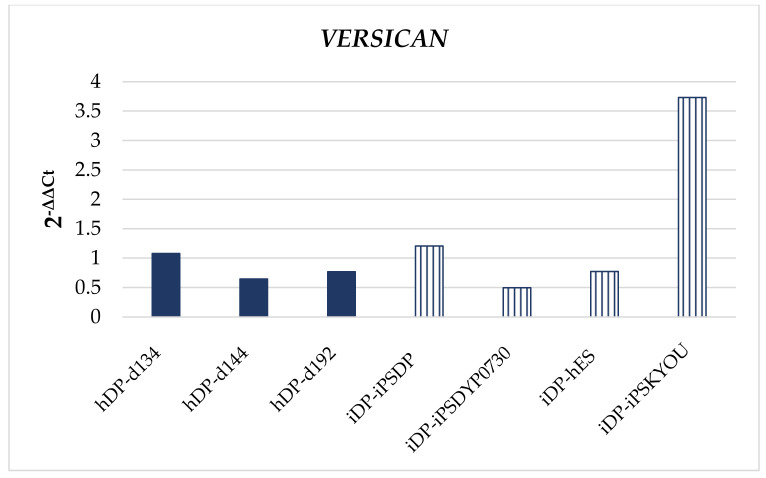
Expression of specialized markers in the dermal spheroids. Quantitative PCR analysis: *VERSICAN*, *ALCALINE PHOSPHOTASE* (*ALP*), *VIMENTIN*, *FIBRONECTIN*, *NESTIN*, *COLLAGEN I*, *COLLAGEN III*.

**Figure 6 cells-11-02730-f006:**
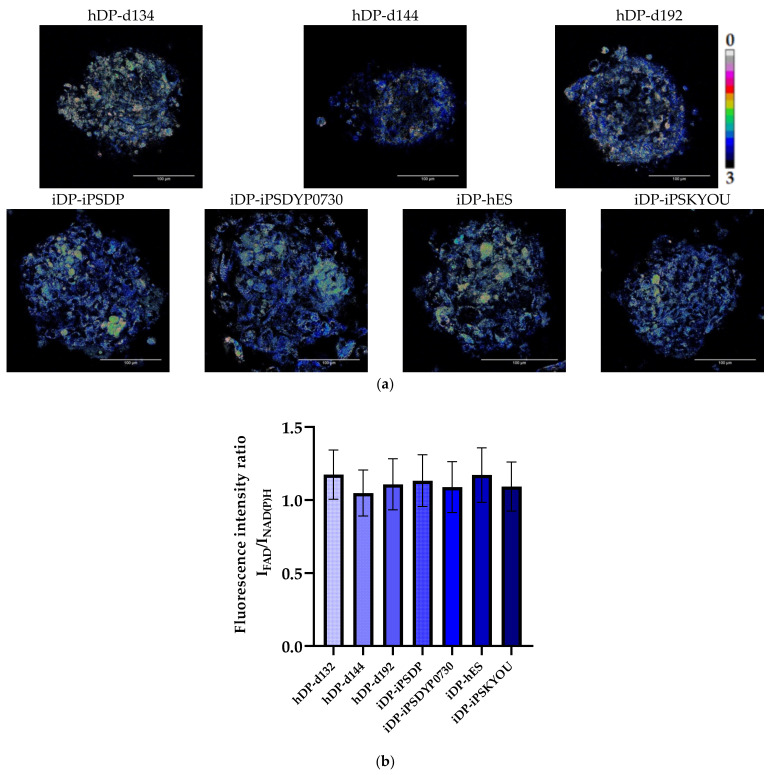
ORR analysis of the hDP spheroids, created from hDP-d134, hDP-d144, and hDP-d192 lines, and iDP spheroids, created from iDP-iPSDP, iDP-iPSDYP0730, iDP-hES and iDP-iPSKYOU lines. (**a**) Images of the fluorescence intensity ratio FAD to NAD(P)H in hDP and iDP spheroids (the scale length in all pictures is 100 µm). (**b**) Values of the fluorescence intensity ratio FAD to NAD(P)H in hDP and iDP spheroids (mean ± SD).

**Figure 7 cells-11-02730-f007:**
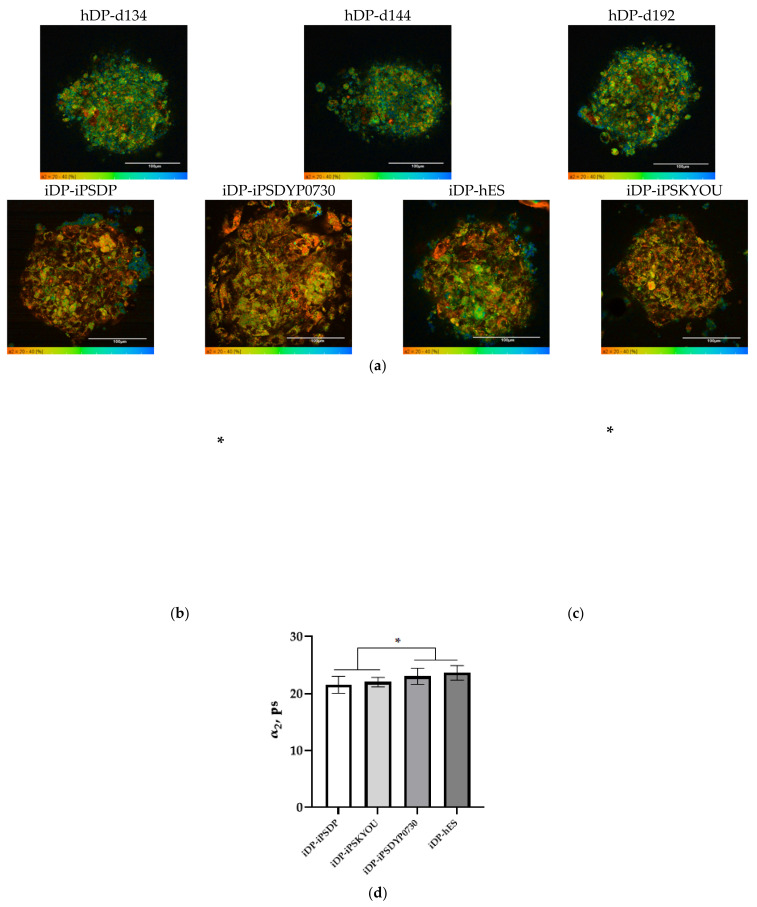
FLIM of NAD(P)H in the hDP spheroids, created from hDP-d134, hDP-d144, and hDP-d192 lines, and iDP spheroids, created from iDP-iPSDP, iDP-iPSDYP0730, iDP-hES, and iDP-iPSKYOU lines. (**a**) Pseudocolor-coded images of the contribution of the bound form of NAD(P)H in hDP and iDP spheroids (the scale bar for α2 on images from 20% to 40%, the scale length in all pictures is 100 µm). (**b**) Mean fluorescence lifetime of NAD(P)H in hDP and iDP spheroids (mean ± SD). (**c**) Fluorescence lifetime contributions of the bound form of NAD(P)H in hDP and iDP spheroids (mean ± SD). (**d**) Fluorescence lifetime contributions of the bound form of NAD(P)H in iDP spheroids created from different cell lines (mean ± SD). * *p*-value ≤ 0.05.

**Figure 8 cells-11-02730-f008:**
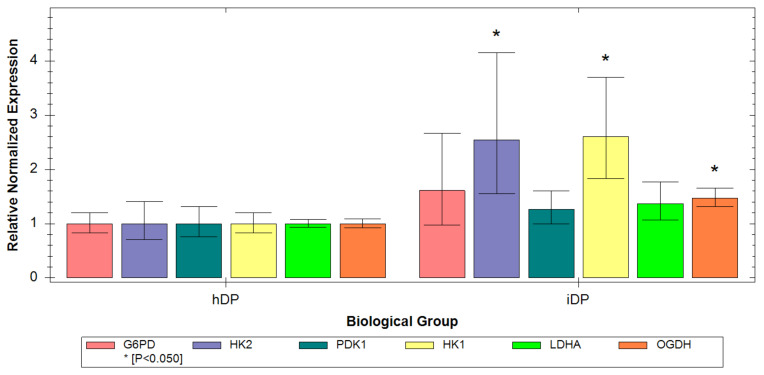
Expression of metabolic markers in the dermal spheroids. Quantitative PCR analysis: G6PD, HK2, PDK1, HK1, LDHA, and OGDH. Data were obtained using 3–4 independent biological replicates and are presented as mean *±* SD.

**Figure 9 cells-11-02730-f009:**
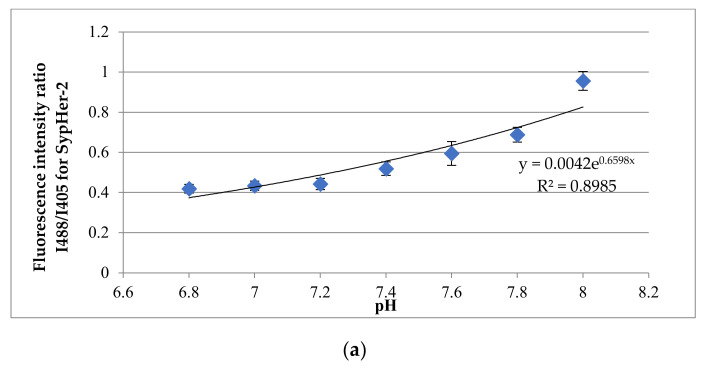
Analysis of the intracellular pH in iDP and hDP spheroids. (**a**) Calibration curve of the dependence of the ratio of fluorescence intensities on the pH for SypHer-2 in iDP spheroids (mean ± SD); (**b**) calibration curve of the dependence of the ratio of fluorescence intensities on the pH for BCECF in hDP spheroids (mean ± SD).

**Figure 10 cells-11-02730-f010:**
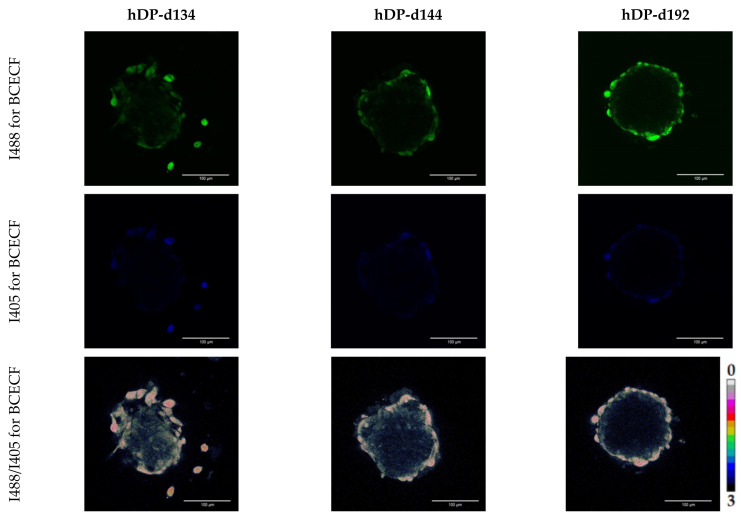
Analysis of the intracellular pH hDP spheroids: Images of the fluorescence intensity of BCECF when excited at a wavelength of 488 nm and 405 nm and images of the fluorescence intensity ratio I488/I405 in hDP spheroids. Excitation of BCECF fluorescence at 488 and 405 nm, fluorescence detection at 500–550 nm. Images size is 213 × 213 μm, the scale length in all pictures is 100 µm.

**Figure 11 cells-11-02730-f011:**
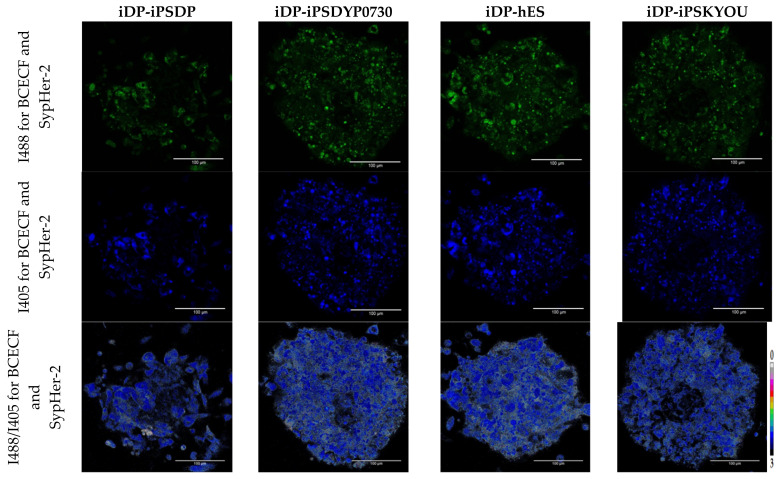
Analysis of the intracellular pH iDP spheroids: Images of the fluorescence intensity of SypHer-2 and BCECF when excited at a wavelength of 488 nm and 405 nm and images of the fluorescence intensity ratio I488/I405 in iDP spheroids. Excitation of SypHer-2 and BCECF fluorescence at 488 and 405 nm, fluorescence detection at 500–550 nm. Images size is 213 × 213 μm, the scale length in all pictures is 100 µm.

**Table 1 cells-11-02730-t001:** The primer sequences for RT-PCR analysis of the expression of specialized genes.

Primer Target	Primer Sequence (5′→3′)
*OCT4*	Forward Primer-ACCCACACTGCAGCAGATCA
	Reverse Primer-CACACTCGGACCACATCCTTCT
*SOX2*	Forward Primer-TGCGAGCGCTGCACAT
	Reverse Primer-GCAGCGTGTACTTATCCTTCTTCA
*NANOG*	Forward Primer-GTCTCGTATTTGCTGCATCGT
	Reverse Primer-AACACTCGGTGAAATCAGGGT
*TUBB3*	Forward Primer-FW CAACAGCGACGGAGGTCTC
	Reverse Primer-AAGACAGAGACAGGAGCAGC
*PAX6*	Forward Primer-AGTGCCCGTCCATCTTTGC
	Reverse Primer-CGCTTGGTATGTTATCGTTGGT
*FIBRONECTIN*	Forward Primer-CCAGTTTTGTGACATTCCCTT
	Reverse Primer-GCATTTGCTTATTTCCTTGTG
*VERSICAN*	Forward Primer-TGCCACCCAGTTACAACACC
	Reverse Primer-TGCCACCCAGTTACAACACC
*VIMENTIN*	Forward Primer-GATGTTTCCAAGCCTGACCT
	Reverse Primer-TACCATTCTTCTGCCTCCTG
*NESTIN*	Forward Primer-GAGAAACAGGGCCTACAGAGC
	Reverse Primer-GGCTGAGGGACATCTTGAGG
*ALKALINE PHOSPHATASE*	Forward Primer-ATGAGGCGGTGGAGATGGAC
	Reverse Primer-AATGTGAAGACGTGGGAATGGT
*Collagen I*	Forward Primer-AGAAAGGGGTCTCCATGGTG
	Reverse Primer-AGGACCTCGGCTTCCAATAG
*Collagen III*	Forward Primer-CCAGGAGCTAACGGTCTCAG
	Reverse Primer-TGATCCAGGGTTTCCATCTC
*GAPDH*	Forward Primer-CCATGTTCGTCATGGGTGTG
	Reverse Primer-GGTGCTAAGCAGTTGGTGGTG
*PSMB4*	Forward Primer-CATTCCGTCCACTCCCGATT
	Reverse Primer-CGAACTTAACGCCGAGGACT
*REEP5*	Forward Primer-ACTGCATGACTGACCTTCTGG
	Reverse Primer-AGTCCGATGACACCAAGAGC
*C1ORF43*	Forward Primer-ACGCCTTTCAAGGGTGTACG
	Reverse Primer-CAAAGACCCCTGTCCCATAGC
*ACTB*	Forward Primer-TGCGTTGTTACAGGAAGTCCC
	Reverse Primer-GCTATCACCTCCCCTGTGTG

**Table 2 cells-11-02730-t002:** The primer sequences for RT-PCR analysis of the expression of metabolic genes.

Primer Target	Primer Sequence (5’→3’)
*HK1*	Forward Primer-CTGCTGGTGAAAATCCGTAGTGG
	Reverse Primer-GTCCAAGAAGTCAGAGATGCAGG
*HK2*	Forward Primer-GAGTTTGACCTGGATGTGGTTGC
	Reverse Primer-CCTCCATGTAGCAGGCATTGCT
*PDK1*	Forward Primer-CATGTCACGCTGGGTAATGAGG
	Reverse Primer-CTCAACACGAGGTCTTGGTGCA
*LDHA*	Forward Primer-GGATCTCCAACATGGCAGCCTT
	Reverse Primer-AGACGGCTTTCTCCCTCTTGCT
*G6PD*	Forward Primer-CTGTTCCGTGAGGACCAGATCT
	Reverse Primer-TGAAGGTGAGGATAACGCAGGC
*OGDH*	Forward Primer-GAGGCTGTCATGTACGTGTGCA
	Reverse Primer-TACATGAGCGGCTGCGTGAACA
*ABL1*	Forward Primer-CCAGGTGTATGAGCTGCTAGAG
	Reverse Primer-GTCAGAGGGATTCCACTGCCAA
*EIF2B*	Forward Primer-CTACTCCAGAGTGGTCCTGAGA
	Reverse Primer-GTTGAGGTGGCAGAGGGCTTTG

**Table 3 cells-11-02730-t003:** Data on the redox ratio and FLIM parameters in hDP and iDP spheroids (mean ± SD).

Type of Spheroids		Redox Ratio	τm (ps)	τ1 (ps)	τ2 (ps)	α2, %
hDP	hDP-d134	1.17 ± 0.17	840.58 ± 48.83	324.85 ± 31.34	2169.36 ± 78.82	27.77± 1.31
	hDP-d144	1.05 ± 0.16	807.24 ± 41.02	318.72 ± 23.77	2147.32 ± 90.50	27.09 ± 1.22
	hDP-d192	1.11 ± 0.17	795.98 ± 31.42	322.29 ± 27.29	2124.53 ± 75.41	26.46 ± 1.03
iDP	iDP-iPSDP	1.13 ± 0.18	628.96 ± 84.35	257.28 ± 42.79	2062.52 ± 185.35	21.43 ± 1.70
	iDP-iPSDYP0730	1.09 ± 0.17	664.74 ± 76.88	259.73 ± 41.14	2016.94 ± 116.72	23.02 ± 1.41
	iDP-hES	1.17 ± 0.19	714.27 ± 62.76	292.66 ± 29.67	2068.76 ± 67.66	23.63 ± 1.28
	iDP-iPSKYOU	1.09 ± 0.17	670.92 ± 54.01	279.09 ± 28.34	2028.40 ± 79.29	22.42 ± 1.00

## Data Availability

The data presented in this study are available on request from the corresponding author.

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
