# Peer review of "FLIM for Evaluation of Difference in Metabolic Status between Native and Differentiated from iPSCs Dermal Papilla Cells"

_cells, 2022, doi:10.3390/cells11172730_

Round 1
Reviewer 1 Report
The manuscript by Kashirina et al describes the possible application of FLIM for evaluation of difference in metabolic status between native dermal papilla cells and those differentiated from iPSCs in 3D models. There is a definite need to develop new approaches and techniques such as dermal papilla cells used to create skin appendages, rather than simply composition of primary keratinocytes and fibroblasts. The aim of this work is to study and compare the metabolic status and pH of dermal spheroids created from dermal papilla cells differentiated from iPSC and native dermal papilla cells using fluorescence microscopy and FLIM. They compared 3D cultures from DP cells differentiated from iPSCs to native human cell models by analyzing energy metabolism and intracellular pH using minimally invasive fluorescent and FLIM methods. However, the quality of iDP cells and their ability to aggregate may depends on the type of iPSCs, and different types of iPSC should be considered in this study. In addition, there are few other gene phenotypes in the dermal spheroids during different stages except those as listed in this article. Due to this concern as well as those delineated below, it is recommended that the manuscript undergo a major revision before reconsideration for publication.
Scientific aspects:
- Dermal papilla cells differentiated from the iPSCs need more induction basis in method section. (Line 110-113)"Further differentiation was induced using DMEM/F12 medium supplemented with insulin/transferrin/selenite (ITS) (Gibco, Thermo Fisher Scientific, Waltham, MA, USA) 10% FBS (HyClone) and gentamicin / amphotericin (Gibco, Thermo Fisher Scientific, Waltham, MA, USA) (1/500). After 3 weeks, the cell culture was transferred to an atmosphere of 21% O2. "
The scheme is not specific enough, so it is suggested to use a flow chart in the Fig1. In addition, dermal papilla cannot be clearly seen only from the current differentiation, which may need more supporting data. Without additional confirmation, the conclusion based on the dermal spheroids created from dermal papilla cells differentiated from iPSC is highly correlative but too weak.
- Only one kind of iPS cell line (iPSC-KYOU line) was selected for induction in this article and the induced dermal spheroid morphology was poor in Fig1. Please refer to the following protocol: "Lee J, van der Valk WH, Serdy SA, Deakin C, Kim J, Le AP, Koehler KR. Generation and characterization of hair-bearing skin organoids from human pluripotent stem cells. Nat Protoc. 2022 Mar 23. doi: 10.1038/s41596-022-00681-y. Epub ahead of print. PMID: 35322210." They selected different kinds of iPS cells to reach their conclusions. In addition, three hair follicles aged 41, 57 and 60 (Line 116-117) were selected. Did tissue samples from different ages have an impact on the subsequent metabolic results?
- This study lacks the comparative data of 2D and 3D, and directly obtains good results of 3D. Beyond that the methods in this study of metabolism, what is the difference and how do they differ from the below article " Rodimova SA, Meleshina AV, Kalabusheva EP, Dashinimaev EB, Reunov DG, Torgomyan HG, Vorotelyak EA, Zagaynova EV. Metabolic activity and intracellular pH in induced pluripotent stem cells differentiating in dermal and epidermal directions. Methods Appl Fluoresc. 2019 Sep 9;7(4):044002. doi: 10.1088/2050-6120/ab3b3d. PMID: 31412329. Please state the innovation of your study and improvement.
Minor aspects:
- The article title is too long and FLIM is not well explained in its first appearance.
- The method section lacks the necessary item catalogue.
- The order of hDP spheroids and iDP spheroids group is not consistent in Fig3 compare with Fig1 and 2.
- Genes related to human being should be capitalized and italicized (line 169-170, Fig2).
- Some of the statements are wrong. For instance:
1)The first letter should be lowercase, iPS(line 97,100,107)
2)Lack of expression of the first word i,Dermal papilla cells (iDP)(Line 57,97)
3)Line131-132 "containing 104 cells in 20 μL" 104?
Reviewer 2 Report
It is very crucial to establish a new metabolic status detection approach for 3D cell culture in vitro. DP spheroidal culture methods provide a perfect model to analyze the cell energy changes and pH differences. In this manuscript, the metabolism and pH value of two kinds of DP spheroids (hDP, iDP) were compared by fluorescence microscopy and FLIM detection. The authors also characterized the phenotype heterogeneity from these two cells by immunostaining and qPCR for different molecules. Finally, they conclude that the metabolism mode of DP spheres formed by IPSCs differentiation was not spatially heterogeneous, while human cell-derived DP spheres show different metabolic patterns in the central and marginal sites. In general, the biological phenotype described in this paper is novel, but the experimental methods and rationale need to be further improved. Here are some suggestions and questions.
1. It is not sufficient to elucidate the characteristics of metabolism changes in DP cell aggregates only by the ratio of NAD(P)H to FAD, and it is better to characterize it by immunostaining or metabolic kits for key metabolic enzymes.
2. In figure1 b and 1c, the expression of red fluorescence is barely visible.
3. Line 85, you mentioned “More alkaline pH values are known to promote glycolysis in cells”. Can you prove this through your data or others’ research data?
4. Line 133, there is no medium added during the first three days of cell culture, will this affect their phenotype?
5. In Figures 4c and 4d, you should show all pH pictures from two groups of DP cells under two detection probes to help readers make a comparative analysis.
6. The title of the article needs revision.
7. Please improve the quality of some fluorescent images.
8. In this research, the cultured DP cells were derived from 3 human donors. Is there any difference between their DP spheroids?
Round 2
Reviewer 1 Report
Overall, the authors did a nice job addressing my initial concerns. However, there are a couple of concerns about the interpretation of some of the new data. In addition, some of my criticism regarding this work still stands.
1. “The scheme is not specific enough, so it is suggested to use a flow chart in the Fig1.” No changes were made to the revised manuscript for this comment.
2. Time point of gene changes in the differentiation to DP is too few and additional investigations about the markers related to early-differentiation and mid-differentiation are necessary in Fig1 and Fig2.
3. Gene symbol related to human being should be capitalized and italicized in Table1, Table2 and Fig 2. No changes were made to the revised manuscript for this comment.
4. Were all the experiments in Figure 2 carried out independently? Please specify the number of repetitions of the experiment.
5. Some of the statements are still wrong. For instance:
1)The first letter should be lowercase, “IPS”(Line116)
2)”iPSCs and hESCs “was” preliminarily” (Line 122-123)
3)Catalogue numbers are preferred to be stated. (Line168-170, Line172)
4) Figure 3、4、7、8 are missing the scale bars for the images.
